# SPATIAL GENERALIZATION OF VISUAL IMITATION LEARNING WITH POSITION-INVARIANT REGULARIZATION

## ABSTRACT

How the visual imitation learning models can generalize to novel unseen visual observations is a highly challenging problem. Such a generalization ability is very crucial for their real-world applications. Since this generalization problem has many different aspects, we focus on one case called *spatial generalization*, which refers to generalization to unseen setup of object (entity) locations in a task, such as a novel setup of object locations in the robotic manipulation problem. In this case, previous works observe that the visual imitation learning models will overfit to the absolute information (e.g., coordinates) rather than the relational information between objects, which is more important for decision making. As a result, the models will perform poorly in novel object location setups. Nevertheless, so far, it remains unclear how we can solve this problem effectively. Our insight into this problem is to explicitly remove the absolute information from the features learned by imitation learning models so that the models can use robust, relational information to make decisions. To this end, we propose a novel, position-invariant regularizer for generalization. The proposed regularizer will penalize the imitation learning model when its features contain absolute, positional information of objects. We carry out experiments on the MAGICAL and ProcGen benchmark, as well as a real-world robot manipulation problem. We find that our regularizer can effectively boost the spatial generalization performance of imitation learning models. Through both qualitative and quantitative analysis, we verify that our method does learn robust relational representations.

## 1 INTRODUCTION

Imitation learning is a class of algorithms that enable robots to acquire behaviors from human demonstrations (Hussein et al., 2017). The recent advance in deep learning has boosted the development of visual imitation learning and supported its applications like autonomous driving, robotic manipulation, and human-robot interaction (Hussein et al., 2017).

In spite of its success, visual imitation learning methods still face many practical challenges. One major challenge is its ability to generalize to novel unseen visual observations, which is very common when we deploy the trained models (Toyer et al., 2020; Park et al., 2021). In the literature, this generalization problem is also known as the robustness problem. The problem covers many different aspects. For example, here we can identify two basic generalization capabilities: *observational generalization* and *spatial generalization* (Figure 1). Observational generalization refers to the generalization to novel visual textures. The changes in background color, object texture, or ambient light in the robotic manipulation task are examples of observational generalization. Such kind of visual change does not affect the physics structure (e.g., the position of object and targets) and only requires the robot to reason about semantic meanings correctly. In contrast, spatial generalization refers to the generalization to unseen setup of objects' (entities) locations in one task, which instead requires physical common sense about space and object. Consider the task of letting a warehouse robot move a box to some target region. If we set the initial position of the box to a place that is not covered by the demonstration dataset, then the imitation learning methods must be able to perform spatial generalization so as to succeed. In reality, the generalization challenge usually emerges as a

Figure 1: **Left and Middle:** Two kinds of visual generalization. The examples are based on the MAGICAL benchmark provided by Toyer et al. (2020), in which a robot is required to relocate a box to a target region. The left figure shows an example of observational generalization, in which the only change during the testing phase is the visual texture of objects. The middle figure shows an example of spatial generalization. The object setup in the testing phase is unseen. **Right:** To achieve spatial generalization, we suggest that absolute information should be removed from the feature while the relational information should be kept. We propose a novel, position-invariant regularizer for this purpose.

combination of different generalization capabilities. In this paper, *we focus on the study of spatial generalization*.

For better spatial generalization, the visual imitation learning models should be able to obtain knowledge about objects and their spatial relations with proper inductive biases. Some work finds that vanilla deep visual imitation learning models strongly overfit to the absolute position of objects (Toyer et al., 2020), which suggests that they do not extract relational information of objects to make decisions like humans (Doumas et al., 2022). Since the representation learning methods can usually lead to good semantic representations (features) and lead to generalization, Chen et al. (2021) investigate the use of self-supervised representation learning methods in visual imitation learning. However, they find that these general-purpose representations fail to improve the generalization performance of vanilla visual imitation learning models effectively. Aside from these works, we also notice that some works propose variants of vision transformers (Dosovitskiy et al., 2021) to improve spatial generalization (Yuan et al., 2021), though these methods are not for imitation learning. Moreover, they make additional assumptions, such as the availability of object information. So far, it remains unclear how to ensure spatial generalization in visual imitation learning.

Based on these observations, our main insight into this problem is to explicitly remove the absolute, positional information from the learned features in the visual imitation learning models. Note that this does not mean that the decision-making process is not dependent on absolute information. Rather, we expect that the model can extract the relational information (e.g., distance, direction) from the absolute information to make robust decisions. To this end, we propose a novel position-invariant regularizer called POINT. This regularizer will penalize the imitation learning model when it finds that the learned feature highly correlates with absolute, positional information. As a result, the imitation learning model has to discover more robust relational features. To validate our idea, we test the proposed regularizer on the MAGICAL (Toyer et al., 2020) and ProcGen (Cobbe et al., 2020) benchmark, as well as a real-world robot manipulation problem. We find that our method can effectively improve spatial generalization performance. Furthermore, we conduct qualitative and quantitative analysis and find that the imitation learning models can indeed learn relational features with our proposed regularizer.

To summarize, our contributions in this paper are as follows.

- We define the spatial generalization problem of visual imitation learning models and propose a novel position-invariant regularizer called POINT to tackle this problem.
- We test our method on MAGICAL and ProcGen benchmarks, as well as a real-world robot manipulation problem. We find that our proposed regularizer can improve the spatial generalization performance of previous imitation learning models effectively.
- Through qualitative and quantitative studies, we verify that our proposed regularizer does make the visual imitation learning models extract relational information.

## 2 RELATED WORK

**Imitation Learning**   Imitation learning (IL) is a classical method to solve sequential decision-making problems with expert demonstration data. Existing imitation learning methods consist of two classes of algorithms: behavioral cloning (Bain & Sammut, 1995) and inverse reinforcement learning (Ng et al., 2000). Behavioral cloning is a supervised learning method that directly fits experts' actions. Inverse reinforcement learning instead proposes to infer a reward function from experts' demonstrations and use it to train an RL agent. Both of these methods suffer from the generalization problem. In this paper, we mainly focus on behavioral cloning since it is simple yet effective in visual domains, and does not involve dangerous online interactions like IRL (Park et al., 2021).

**Generalizable Policy Learning**   The generalization problem of imitation learning has been a long-standing problem. Since the problem also exists in reinforcement learning and most of the solutions can be adapted to the case of imitation, we discuss the works from both fields here. One existing branch of work is the *domain randomization* (Tobin et al., 2017). The basic idea of domain randomization is to collect data from diverse setups, such as different backgrounds and textures. As a result, the trained model will be able to discover more robust features. Resembling this idea, another line of work tries to solve this problem with *data augmentation* (Yarats et al., 2020; Hansen & Wang, 2021), which augments the input or the learned features according to some heuristics. For example, MixReg (Wang et al., 2020) proposes a mixture augmentation method and leads to better out-of-distribution generalization. OREO (Park et al., 2021) proposes to mask out the objects in the feature map so that the model will not fit a specific object in the observation. CLOP (Bertoin & Rachelson, 2022) finds it important to shuffle the feature map locally to ensure robustness. Though not exactly a data augmentation approach, De Haan et al. (2019) propose to randomly mask out the semantic components in the learned feature and check whether they are correlated with correct expert actions. Aside from works using diverse data or data augmentation, some works also approach this problem by incorporating proper inductive biases into the design of policy networks (Dasari & Gupta, 2020). For instance, SORNet (Yuan et al., 2021) uses a variant of vision-transformer to solve robotics tasks that require relational reasoning. However, SORNet is not aimed at imitation learning or reinforcement learning. Zhou et al. (2022) propose to use an attentional architecture to solve a compositional, object-centric reinforcement learning problem. The idea of these methods is to leverage the relational inductive bias of the attention operations. Wen et al. (2022) propose PrimeNet, which introduces a shortcut into the neural network to ensure that the model focus on more meaningful areas. Representation learning methods are another approach for generalization. Some works use pretrained representation. For instance, Yen-Chen et al. (2020) use a pretrained model on MS-COCO dataset (Lin et al., 2014) to provide representation for imitation learning in robotics manipulation. The pretrained model over large datasets provides more robust object representations and can generalize the policy to unseen objects. Besides pretrained representation, self-supervised representation learning (Mandi et al., 2022; Chen et al., 2021) on the demonstration dataset can also improve the generalization in some cases. However, Chen et al. (2021) suggest that self-supervised representations do not improve the generalization performance in general.

We refer readers to (Kirk et al., 2021) for comprehensive knowledge of this field. Our work differentiates from all these existing works by explicitly defining the spatial generalization problem and proposing to remove the nonrobust positional information from the representation.

## 3 PRELIMINARIES

**Notations**   We model the sequential decision making problem as a Markov Decision Process $\mathcal{M} = (\mathcal{S}, \mathcal{A}, \mathcal{R}, \mathcal{T})$. $\mathcal{S}$ is the state space. $\mathcal{A}$ is the action space. $\mathcal{R}$ is the reward function. $\mathcal{T}$ is the transition dynamics. The agent's state at timestep $t$ is $s_t \in \mathcal{S}$. The agent takes action $a_t$ and receives reward $r_t = \mathcal{R}(s_t, a_t)$. Its state at timestep $t + 1$ is then $s_{t+1} \sim \mathcal{T}(s_t, a_t)$. The objective of the agent is to maximize the return $\sum_{t=0}^{T} \gamma^t r_t$, where $\gamma \in (0, 1]$ is a discount factor.

For the imitation learning problem studied here, the agent has no access to $\mathcal{R}$ and $\mathcal{T}$, but it is provided with a fixed expert demonstration dataset $\mathcal{D} = \{\tau_i\}$. Here, each $\tau_i = (s_0^E, a_0^E, s_1^E, a_1^E, ...s_T^E, a_T^E)$ is an expert trajectory that can achieve high performance (return) in $\mathcal{M}$. Therefore, the agent should learn the behavior by leveraging the given demonstration dataset.

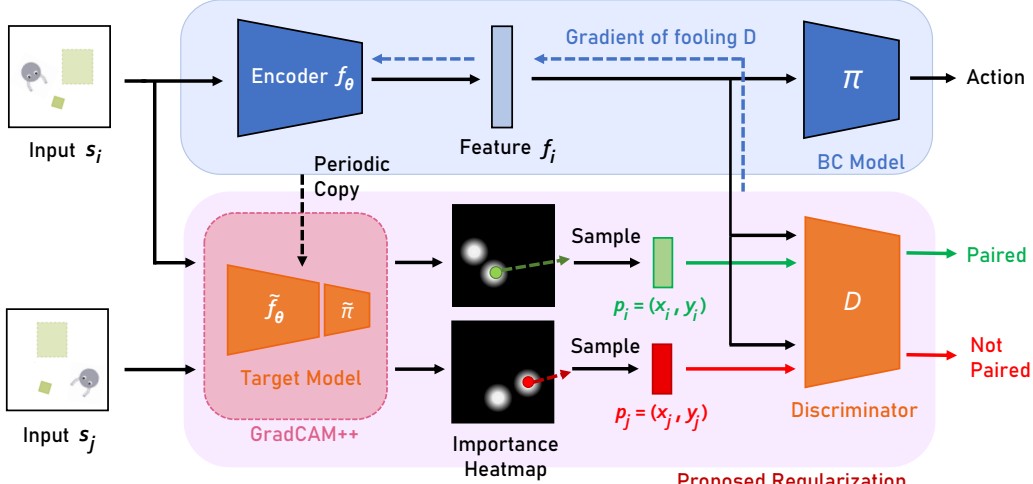

Figure 2: Overview of our method. The blue branch above is the common imitation learning (BC) pipeline. Our proposed regularizer is shown in the light pink box at the bottom. The regularizer first uses the GradCAM++ algorithm to find out the important areas based on which the latest BC model makes decisions. Then it samples the coordinates from the discovered important areas and trains a discriminator network $D$ to calculate whether these sampled coordinates are paired with the feature $f_i$. The BC model (encoder $f_\theta$) is then trained to fool the discriminator $D$. When the encoder $f_\theta$ is able to fool $D$, the absolute positional information is removed from the feature as desired.

**Behavioral Cloning**    One classical imitation learning algorithm is the Behavioral Cloning (BC). BC turns the imitation learning problem into a supervised learning problem. It fits the expert's action $a_i$ given the observation $s_i$. For the visual imitation learning problem, the BC model can be divided into two consecutive parts: a vision encoder $f_\theta$ (which is usually a convolutional neural network), and a policy head $\pi$. The $f_\theta$ first encodes $s_i$ to the feature $f_i = f_\theta(s_i)$, and the $\pi$ then uses it to predict the expert's action. The BC algorithm minimizes the following negative log-likelihood objective:

$$\mathcal{L}_{BC} = \mathbb{E}_{(s_i, a_i) \in \mathcal{D}} \left[ - \log \pi(a_i | f_\theta(s_i)) \right]. \tag{1}$$

Due to its simplicity, BC is widely used in visual imitation learning. Therefore, we study the spatial generalization of BC in this paper.

**GradCAM**    Gradient-weighted Class Activation Mapping (GradCAM) (Selvaraju et al., 2017; Chattopadhay et al., 2018) is a class of algorithms that can interpret the deep vision models by visualizing the important region for decision making. Its basic idea is to calculate the gradient at each feature map and aggregate them into an importance map. In this paper, we leverage GradCAM as a rough object detector in the proposed regularizer. We refer readers to the original paper for more technical details.

## 4 METHOD

In this section, we present our position-invariant regularizer for spatial generalization. As is discussed before, our basic idea is to remove the absolute positional information from the features of the BC model. In the following parts, we first present a formalized description of the idea and its practical challenges in Section 4.1. Then, we discuss how to handle these challenges with GradCAM and adversarial training in Section 4.2 and 4.3. Finally, we provide a summary of the overall algorithm in Section 4.4. We illustrate our method in Figure 2 and provide the pseudo code in Algorithm 1.

### 4.1 FORMULATION AND CHALLENGES

For the tasks that involve spatial generalization, there usually exist multiple objects in the observed states, such as the agent, the target object, and the goal. For the state $s_i$, we denote each of these objects

in $s_i$ as $o_i^j$, and their positions as $(x_i^j, y_i^j)$. Then, our idea can be formulated as the minimization problem of each $I((\mathbf{x}^j, \mathbf{y}^j), \mathbf{f})$, where $I$ is the mutual information. Note that we use the notation $\mathbf{x}^j, \mathbf{y}^j, \mathbf{f}$ to indicate the corresponding random variables of $x_i^j, y_i^j, f_i$. However, this formulation leads to many practical challenges. First, since each $(x_i^j, y_i^j)$ is not provided directly by $s_i$ and should be inferred, we have to either train some object key-point detectors to detect the underlying objects in the training set, or annotate the objects by ourselves. However, both of these approaches can be difficult and tedious in practice. Second, even if we have ideal key-point detectors, we have to deal with a hard optimization problem in the summation form $\sum_j I((\mathbf{x}^j, \mathbf{y}^j), \mathbf{f})$. This can be intractable when there are many objects in the observed state.

Fortunately, we find that the previous works on the interpretation of deep learning models like GradCAM provide useful tools to handle these challenges. It can reduce the problem to a much simpler form. We discuss our observations as follows.

## 4.2 PROBLEM REDUCTION WITH GRADCAM

GradCAM is an interpretation method that can tell which part of the image is crucial in the decision process of a deep learning model. Given a BC model $(f_\theta, \pi)$ and input $s$, GradCAM outputs an importance heatmap of the same resolution as the input $s$. The heatmap indicates the importance of each pixel when we use this BC model for prediction. One nice property of this generated heatmap is that it is smooth and usually coincides with the meaningful objects in the input $s$. Therefore, we can consider the GradCAM as a rough object detector here.

We propose to sample $p_i = (x_i, y_i)$ from the generated heatmap, and then minimize the $I(\mathbf{p}, \mathbf{f})$. We find that this new objective can act as a proxy for the original objective in practice. Concretely, if $p_i$ is always far from a specific object like $o^k$, then we know that $o^k$ is irrelevant to the decision process of the current model. In this case, we conjecture that $I((\mathbf{x}^k, \mathbf{y}^k), \mathbf{f})$ should be low enough to meet our requirement. On the contrary, if $p_i$ always coincides with a certain object like $o^l$, then we actually minimize $I(\mathbf{p}, \mathbf{f}) \approx I((\mathbf{x}^l, \mathbf{y}^l), \mathbf{f})$ as we want.

## 4.3 LOSS FUNCTIONS

Now, our remaining work is to reduce the mutual information $I(\mathbf{p}, \mathbf{f})$. However, we find that jointly estimating and minimizing the mutual information in our vision-based tasks is hard in practice. Since our ultimate goal is to minimize the information of $\mathbf{p}$ in $\mathbf{f}$, we instead propose an adversarial training framework to achieve this goal.

Specifically, we introduce a discriminator network $D$ to play a two-player min-max game with the BC model as follows.

$$\min_{f_\theta} \max_D \mathbb{E}_{(s_i, a_i) \sim \mathcal{D}, (s_j, a_j) \sim \mathcal{D}} \left[ \log D(p_i, f_i) + \log(1 - D(p_j, f_i)) \right]. \tag{2}$$

In this min-max game, the discriminator $D$ tries to tell the joint distribution of $\mathbf{p}$ and $\mathbf{f}$, denoted as $\mathbb{P}_{\mathbf{p}, \mathbf{f}}$, from the product of their marginal distributions $\mathbb{P}_{\mathbf{p} \otimes \mathbf{f}}$. Meanwhile, the BC model is trying to fool the discriminator by removing the information of $\mathbf{p}$ from $\mathbf{f}$. Applying the convergence theory of the generative adversarial network (GAN) (Goodfellow et al., 2020), we know that when $f_\theta$ is a global minimizer of Equation 2, $\mathbb{P}_{\mathbf{p}, \mathbf{f}} = \mathbb{P}_{\mathbf{p} \otimes \mathbf{f}}$, which implies that $I(\mathbf{p}, \mathbf{f}) = 0$. Therefore this min-max game fulfills our requirement.

In practice, we train $D$ to minimize the following binary classification loss function:

$$\mathcal{L}_D = -\mathbb{E}_{(s_i, a_i) \sim \mathcal{D}, (s_j, a_j) \sim \mathcal{D}} \left[ \log D(p_i, f_i) + \log(1 - D(p_j, f_i)) \right]. \tag{3}$$

However, for the encoder $f_\theta$, we find that using $-\mathcal{L}_D$ as the loss function for training will result in instabilities. We assume this is because the $f_i$ term is present in both of the two terms in Equation 2, which is different from that in the original GAN objective. Therefore, we propose to use the following loss function for optimization, which we find works well empirically:

$$\mathcal{L}_{reg} = \mathbb{E}_{(s_i, a_i) \sim \mathcal{D}} \left[ \log D(p_i, f_i) \right]. \tag{4}$$

Combining the BC loss, the loss function to train the $f_\theta$ and $\pi$ is then

$$\mathcal{L} = \mathcal{L}_{BC} + \lambda \mathcal{L}_{reg} = \mathbb{E}_{(s_i, a_i) \sim \mathcal{D}} \left[ -\log \pi(a_i | f_\theta(s_i)) + \lambda \log D(p_i, f_i) \right]. \tag{5}$$

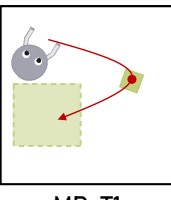 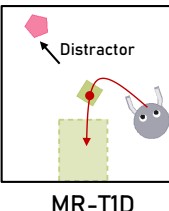 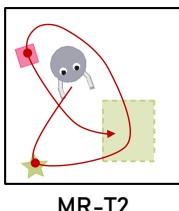 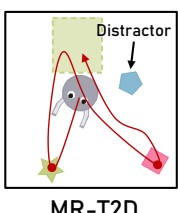 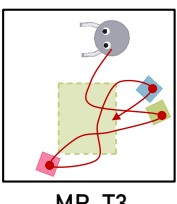

| MR-T1 | MR-T1D | MR-T2 | MR-T2D | MR-T3 |

Figure 3: The MAGICAL tasks used in our experiments. The grey robot is required to move the target objects (we mark them with red dots) to the target region. The red curve shows a possible plan to solve the task (the interaction details like releasing box are omitted). The long horizontal nature of this task brings additional challenges aside from the spatial generalization problem.

### 4.4 SUMMARY OF THE ALGORITHM

Finally, we put everything together and summarize our algorithm in Algorithm 1. We use Grad-CAM++ (Chattopadhay et al., 2018), an improved version of GradCAM, to calculate the heatmap. We find that it can deal with multi-object observation better than GradCAM. Similar to the target network in the deep reinforcement learning, we propose to use a target model in the GradCAM calculation. This target model periodically copies the weights of the latest BC model. This ensures that the $p_i$ terms in Equation 2 will change slowly throughout the training so that the optimization of the min-max game between $D$ and $f_\theta$ becomes stable. Our python implementation of this algorithm is in the supplementary material and Appendix A.1.

---

**Algorithm 1:** POINT

---

Initialize parameters of the BC model: $f_\theta$, $\pi$, and the discriminator $D$. Initialize target model $\tilde{f}_\theta, \tilde{\pi}$ with $f_\theta, \pi$.
**while** *not converged* **do**
    Sample a batch $\{(s_i, a_i)\}$ from $\mathcal{D}$.
    Calculate each importance heatmap
      by $H_i = \text{GradCAM}(\tilde{f}_\theta, \tilde{\pi}, s_i)$.
    Sample each $p_i$ from $H_i$.
    Optimize the BC model by Eqn. 5.
    Optimize the discriminator by Eqn. 3.
    Periodically update $\tilde{f}_\theta, \tilde{\pi}$ with $f_\theta, \pi$.

---

## 5 EXPERIMENTS

In the experiments, we first test the performance of our method on two benchmarks that require spatial generalization: MAGICAL and ProcGen. We study the generalization according to the IID protocol (Kirk et al., 2021). This means that the training and testing task distributions are the same, though the test instance will be unseen. Then, we provide an analysis of our algorithm through both qualitative and quantitive studies. Finally, we extend our method to a real robot manipulation problem.

### 5.1 TASK SETUP

**MAGICAL** The MAGICAL benchmark simulates a 2D robotic manipulation problem in a warehouse room. The tasks provided by the MAGICAL involve complex interactions between the agent and multiple objects, which require effective spatial generalization..

In the experiments, we use a variant of its MatchRegion task. In this task, a robot is required to go across a square room to move some objects to a target region specified by a dashed rectangle. We set up several task instances of the MatchRegion task: MatchRegion-Target-1, MatchRegion-Target-2, MatchRegion-Target-2-Distract, MatchRegion-Target-3, MatchRegion-Target-3-Distract. We provide an illustration of these tasks in Figure 3. For each MatchRegion-Target-$X$ task (MR-T$X$), there is no distractor object in the room, so the robot only needs to move all the $X$ objects into the target location. However, for the MatchRegion-Target-$X$-Distract task (MR-T$X$D), there is an additional distractor object in the room. This object is also randomly placed in the room during testing. The existence of this distractor object not only increases the risk of learning spurious features but also adds to the difficulty of learning secure motions. As we will discuss later, even the existence of one

Table 1: Evaluation result on the MAGICAL and ProcGen benchmark. We show the average score on three random seeds. Our method can achieve state-of-the-art results compared with the baselines.

| Method | Vanilla | Dropout | Crop | Cutout | MixReg | OREO | CLOP | Ours |
|---|---|---|---|---|---|---|---|---|
| MR-T1 | 0.09 ±0.02 | 0.28 ±0.04 | 0.42 ±0.03 | 0.19 ±0.03 | 0.26 ±0.02 | 0.21 ±0.03 | 0.16 ±0.06 | **0.63** ±**0.05** |
| MR-T1D | 0.19 ±0.06 | 0.32 ±0.11 | 0.44 ±0.03 | 0.27 ±0.03 | 0.41 ±0.10 | 0.27 ±0.06 | 0.21 ±0.02 | **0.60** ±**0.08** |
| MR-T2 | 0.25 ±0.03 | 0.48 ±0.03 | 0.46 ±0.04 | 0.43 ±0.05 | 0.44 ±0.05 | 0.37 ±0.05 | 0.32 ±0.07 | **0.75** ±**0.07** |
| MR-T2D | 0.27 ±0.06 | 0.35 ±0.03 | 0.38 ±0.04 | 0.32 ±0.03 | 0.33 ±0.03 | 0.27 ±0.03 | 0.23 ±0.04 | **0.70** ±**0.04** |
| MR-T3 | 0.23 ±0.02 | 0.51 ±0.03 | 0.47 ±0.05 | 0.32 ±0.04 | 0.48 ±0.05 | 0.42 ±0.04 | 0.35 ±0.07 | **0.66** ±**0.03** |
| Coinrun | 0.75 ±0.05 | 0.81 ±0.03 | 0.80 ±0.02 | 0.76 ±0.03 | 0.79 ±0.03 | 0.69 ±0.03 | 0.82 ±0.05 | **0.87** ±**0.02** |
| Jumper | 0.63 ±0.07 | 0.61 ±0.06 | 0.64 ±0.03 | 0.66 ±0.04 | 0.67 ±0.03 | 0.63 ±0.05 | 0.68 ±0.02 | **0.72** ±**0.04** |
| Ninja | 0.65 ±0.06 | 0.69 ±0.02 | 0.71 ±0.02 | 0.72 ±0.02 | 0.68 ±0.02 | 0.67 ±0.01 | 0.67 ±0.02 | **0.79** ±**0.02** |
| Leaper | 0.48 ±0.07 | 0.52 ±0.04 | 0.52 ±0.02 | 0.47 ±0.03 | 0.46 ±0.01 | 0.48 ±0.04 | 0.51 ±0.07 | **0.59** ±**0.05** |
| Miner | 0.16 ±0.07 | 0.28 ±0.05 | 0.25 ±0.02 | 0.21 ±0.07 | 0.23 ±0.03 | 0.22 ±0.02 | 0.19 ±0.04 | **0.35** ±**0.04** |
| Starpilot | 0.33 ±0.05 | 0.53 ±0.14 | 0.61 ±0.08 | 0.39 ±0.06 | 0.55 ±0.10 | 0.35 ±0.03 | 0.42 ±0.04 | **0.71** ±**0.12** |
| Fruitbot | 0.01 ±0.04 | 0.17 ±0.11 | 0.48 ±0.04 | 0.00 ±0.06 | 0.21 ±0.07 | 0.09 ±0.05 | 0.36 ±0.06 | **0.56** ±**0.02** |
| Bigfish | 0.05 ±0.04 | 0.08 ±0.04 | **0.16** ±**0.03** | 0.10 ±0.05 | 0.09 ±0.01 | 0.09 ±0.02 | 0.12 ±0.02 | **0.16** ±**0.04** |

distractor object can lead to a significant increase of generalization difficulty. The study of more distractors is carried out in the analysis part.

For each of the tasks above, we collect its human demonstration dataset by ourselves. For each demonstration trajectory, we randomly set up the initial position of the objects, target region, and the robot. Note that the demonstration dataset provided by the original MAGICAL benchmark uses a fixed initial position setup. However, we find that this setup is too strict for spatial generalization if no other dataset or pretraining tasks are available to provide prior knowledge. For MR-T1, we collect 50 trajectories. For each of the other tasks, we collect 100 trajectories. The collection of all these trajectories takes 2 hours. We also study the outcome of using a different number of trajectories in the later analysis part.

**ProcGen** The ProcGen benchmark is an arcade-game-based benchmark. These arcade games involve interactions between many semantic objects, including the player, monsters (or traps), bonuses, and tools. Though it is designed to test the generalization ability of RL agents, we use it to test the spatial generalization of imitation learning here. We use 8 environments: Coinrun, Jumper, Ninja, Leaper, Miner, Starpilot, Fruitbot, and Bigfish. We follow the 'easy' generalization evaluation protocol used in previous works. We first train a PPO agent (Schulman et al., 2017) to collect demonstrations in the first 200 levels of ProcGen and use these demonstrations to train the imitation learning model. We use 64 expert trajectories in the experiment. Then for the evaluation, we test the imitation learning models at all levels. We remove the background of the game throughout training and testing since this requires observational generalization.

## 5.2 BASELINES

For the vanilla BC policy, we train an IMPALA (Espeholt et al., 2018) policy, whose encoder is a residual convolutional neural network. We also try vision-transformer (Dosovitskiy et al., 2021) and relational network (Santoro et al., 2017) that have relational biases, but we find that they perform worse than IMPALA and do not report their results here. Then, we implement the following baselines for comparison. (1) **Dropout**. Dropout (Srivastava et al., 2014) is a widely-used regularization method in deep learning. Many works also find it useful for improving the performance of imitation learning. (2) **Crop** Crop is a popular data augmentation method used in the state-of-the-art visual RL algorithms like DrQ (Yarats et al., 2020) and RAD (Laskin et al., 2020), it encourages translation

 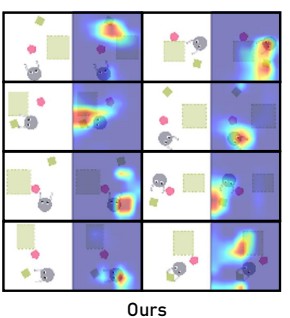

| | Estimated $I((x^i, y^i), \mathbf{f})$ | |
| --- | --- | --- |
| **Object** | **Dropout** | **Ours** |
| | 3.21 | 1.20 |
| | 3.14 | 1.26 |
| | 3.02 | 1.09 |
| | 2.75 | 1.12 |

Dropout   Ours

Figure 4: Left: The GradCAM++ importance heatmap of the dropout model and our model on the MR-T1D task. The red region indicates the most important region, while the dark blue indicates the least important region. The results suggest that the dropout model attends to the red distractor and is not robust. In contrast, our model is able to attend to correct objects. Right: MI between feature and object's position. Our model contains less bits about absolute position information.

invariance to the perspective. (3). **Cutout** Cutout (DeVries & Taylor, 2017) is a data augmentation method. During training, it randomly erases part of the input image. (4). **MixReg** MixReg (Wang et al., 2020) is a data augmentation method. It generates interpolated, synthetic training data based on the original training dataset. (5). **OREO** OREO (Park et al., 2021) is an object-aware data augmentation method. It randomly drops the features of certain objects. (6) **CLOP** (Bertoin & Rachelson, 2022) CLOP is a locality-aware data augmentation method. It randomly shuffles the feature map of the policy network in a small, local region.

## 5.3 RESULTS

**MAGICAL** The result on MAGICAL is shown in Table 1. The performance is defined by the success rate of the trained policy, which is the number of target objects that are successfully transferred to the target region, divided by the total number of target objects. We observe that our method is able to achieve state-of-the-art results and outperform the baselines by a large margin. Concretely, it improves the success rate by about 30%. Besides, we find that most of the previous regularization methods do increase the success rate of the vanilla version and their results are similar to each other. This shows that they may solve some common issues in the generalization problem. However, their performance gap from our method suggests that we tackle a different issue here, which is overfitting to absolute positions.

**ProcGen** The result on ProcGen is shown in Table 1. The performance is calculated as the normalized agent's reward in the test environment (with respect to the random agent's and expert's performance). Though we observe some improvement, the performance gain is smaller in this case compared to that of MAGICAL. We hypothesize that this is because the decision-making process of ProcGen is highly dependent on the local image patches, as suggested by CLOP. In this case, the BC model does not need to perform long-range reasoning across objects frequently, and suffers less from the risk of overfitting to absolute positional information. Our result here also suggests that we need task-specific inductive biases so as to achieve good generalization performance.

## 5.4 ANALYSIS

**Qualitative Results** To understand whether our method is robust, we use GradCAM++ to visualize the importance heatmap of the learned models. For simplicity, we show the result on the MatchReigion-Target-1-Distract task. We compare the result of our model to the model trained with dropout here (Figure 4). We notice that the dropout model tends to focus on the red distractor object rather than the correct target object. In contrast, our model is able to focus on the correct objects. Even when the distance between the agent and the object is large, it can attend to the agent and the object simultaneously. The visualization results suggest that our regularizer indeed leads to relational features even when the vision network IMPALA does not have an explicit relational inductive bias. To understand this, we further use MINE estimator (Belghazi et al., 2018) to measure the MI between

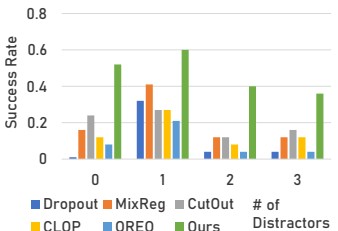

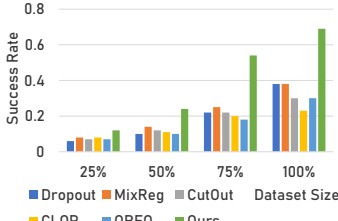

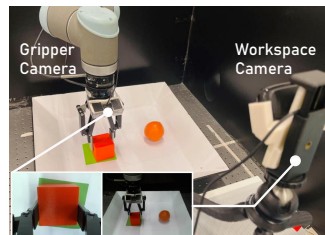

Figure 5: The generalization performance to different number of distractors on MR-T1D.

Figure 6: The variation of performance on MAGICAL using the datasets of different sizes.

Figure 7: The setup of real-world robot manipulation experiments.

the learned representation and the position of different objects. Our representation indeed contains much fewer bits about the absolute position compared with dropout as expected.

**Unseen Number of Distractors**  A robust model should base its decision on robust relational information. As a result, for the MAGICAL tasks, it should be able to ignore the distractor and generalize to an unseen number of distractors. Therefore, we test whether our model trained on MR-T1D (where only one distractor presents) can generalize to MR-T1D with the unseen number of distractors (e.g., 0, 2, 3). We also compare the results with the previous models. The result is shown in Figure 5. We find that our model is able to generalize to the case of 0, 2, 3, though the performance is lower than the case of 1 (training scenario). In contrast, the prior model, such as the dropout model, fails in these unseen cases totally. This echoes our qualitative analysis results.

**Number of Demonstrations**  We also study whether the proposed method works when the amount of expert demonstrations is limited. For this purpose, we test our method on the MAGICAL with $25\%, 50\%, 75\%$ of expert demonstrations. We show the averaged performance in Figure 6. We find that our method can achieve consistent improvement, though the performance decreases as the dataset becomes smaller. This result suggests that we still require a certain amount of diverse data to achieve spatial generalization.

## 5.5   Real-World Experiments

Finally, we test whether our method scales to the real-world pick-and-place manipulation problem. We extend the MR-T1D to a UR10 robot arm with a Robotiq parallel-jaw gripper (Figure 7). As suggested by Hsu et al. (2022), we use a gripper camera and a workspace camera to provide observation. For the BC model, we use two separate IMPALA encoders to process each camera image, concatenate their output features along with the $z$-coordinate of gripper, and feed them into an MLP. We use the proposed regularizer to regularize the workspace branch. We collect 75 human demonstrations for training. We compare our method to dropout with different numbers of distract objects. The result is shown in Table 2. Our method also achieves a large improvement in this problem. We provide detailed setup and qualitative results in the Appendix A.4 and A.3.

Table 2: The success rate of the real-world experiments. Our method is also effective here. Each test consists of 20 trials.

| Method | Dropout | Ours |
|---|---|---|
| 0 Dis. Obj | 35% | **55%** |
| 1 Dis. Obj | 35% | **60%** |
| 2 Dis. Obj | 20% | **50%** |
| 3 Dis. Obj | 10% | **45%** |

## 6   Conclusion

In this paper, we studied the spatial generalization problem of imitation learning. To solve this problem, we proposed a novel position-invariant regularizer to remove the absolute positional information from the features. Through experiments on the MAGICAL and ProcGen benchmarks, as well as a robot manipulation system, we confirmed that previous methods do overfit to the absolute position and showed that our proposed approach can effectively solve this problem. We hope that the proposed method can inspire future generalization research.

## 7 REPRODUCIBILITY STATEMENT

We provide the python source code of the main algorithm and the full model in the supplementary material. We will make our project publicly available.

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

# A  APPENDIX

## A.1  IMPLEMENTATION OF THE ALGORITHM

We provide the core training code of our project in the supplementary material, which comes from our whole project. The code shows all the training details and the model implementation. It is divided into three files.

- `bc.py` This python file provides the main training routine. The main class is the `BehaviorCloning`. The training routine is its `train` method.
- `model.py` This python file provides the implementation of BC policy model (`Model`) and the discriminator (`DiscrimModel`).
- `network.py` This python file provides the implementation of the IMPALA encoder (`ResNetBase`) used by the BC policy model.

Note that these files are for illustration purpose. The whole runnable project will be made publicly available.

## A.2  HYPERPARAMETERS

The size of image observation is $(64, 64)$ in all the experiments. We use the Adam optimizer (Kingma & Ba, 2015) for the optimization of both the discriminator and the BC model. The learning rate of the optimizers is 0.0003. We also add a weight decay of 0.0001 to the BC model. We use a batchsize of 128 throughout the experiments (`batch_size`). We will sample 4 points for each batch (`sample_n`). We repeat the training of discriminator for 4 steps (`repeat`). For the discriminator (`DiscrimModel`), its embedding size is set to 64 (`e_dim`). For the choice of $\lambda$, we use $\lambda = 1.0$ for the MAGICAL, and $0.1$ otherwise. All the other details and hyperparameters can be found in the source code in the supplementary material.

## A.3  QUALITATIVE RESULTS OF THE MANIPULATION PROBLEM

In this section, we provide some qualitative results of the real-world manipulation problem. Recall that in this task, the robot is required to move a red cube to a target location specified by a green area. We show the importance heatmap of the dropout model (Figure 8) and our model (Figure 9). As is shown in the figures, we find that dropout model tends to attend more to the round distractor object compared with our model. However, due to the visual complexity, we find that our model sometimes may attend the shadow in the background.

## A.4  SETUP OF REAL ROBOT

The observation of the robot include two resized $64 \times 64$ RGB images coming from the gripper camera and the workspace camera, and the $z$ coordinate of the gripper. We use a discretized action space, so it will become easier for human to collect the demonstration via the keyboard. There are 8 actions in total, $[x, y, z]$ move [backward/forward] by 2cm, and gripper [open/close] by 1cm. We only consider to regularize the workspace camera. This is because the gripper-camera can only observe a very limited area on the workspace in our case and the gripper is always at the center. As a result, the spatial generalization issue is minor for the gripper-camera and we find it do not help much to regularize it.

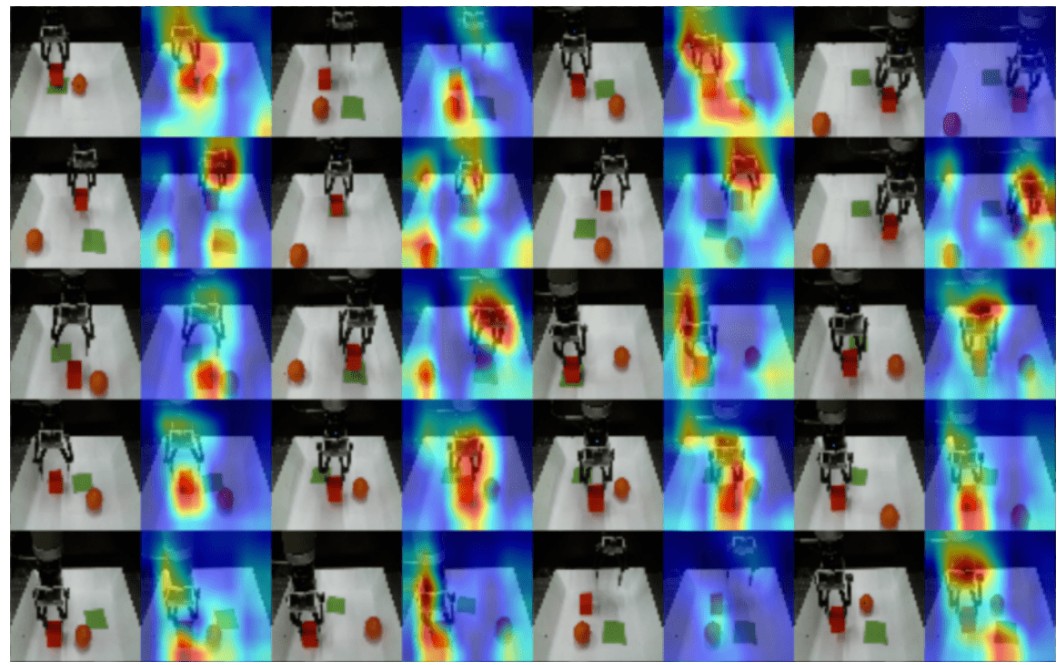

Figure 8: The GradCAM++ importance heatmap of dropout model in the real-world manipulation problem. The dropout model tends to attend the round distractor object.

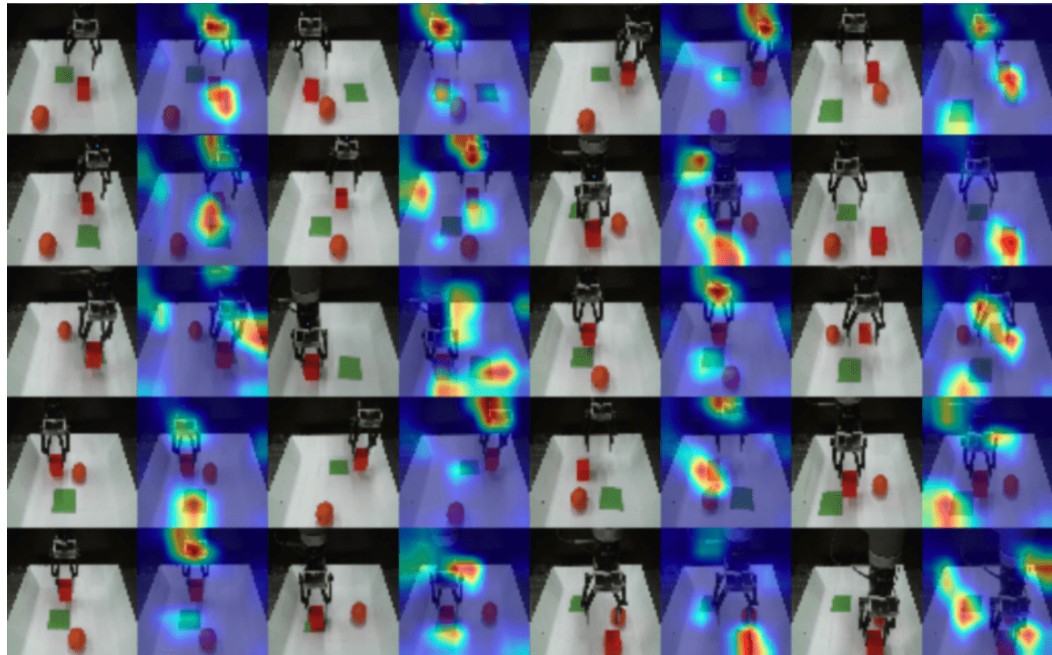

Figure 9: The GradCAM++ importance heatmap of our model in the real-world manipulation problem. Our model attends less to the round distractor object. However, due to the visual complexity, we find that our model sometimes may attend the shadow in the background.

