# OpenReview forum: "Structural Generalization of Visual Imitation Learning with Position-Invariant Regularization"
_ICLR.cc/2023/Conference — Submitted to ICLR 2023_

### Official Review · Reviewer_khtv · 2022-10-20

**Confidence:** 3
**Correctness:** 3
**Technical Novelty And Significance:** 3
**Empirical Novelty And Significance:** 3
**Recommendation:** 5

**Clarity, Quality, Novelty And Reproducibility:**

* Clarity: Good.
* Quality and Novelty: The idea is interesting and the performance, though I do have some concerns (see review above).
* Reproducibility: The authors provide the source code. I would assume the results are reproducible.

**Strength And Weaknesses:**

Strength:
* The paper is in general well written.
* The idea of leveraging GradCAM to handle the challenges (Sec. 4.2) is interesting and clever.

Weaknesses and Comments:
* The experiments on Procgen are a little bit limited. It would be better to evaluate the proposed method on all 16 games.
* Apart from BC, does the position-invariant regularizer also work for other (maybe a SOTA one) imitation learning methods?
* Compared to the baselines, how much extra cost (computation, time) does the adversarial training bring?
* I suggest the authors using the metrics recommended in [1] to compare different methods.

[1] Agarwal, Rishabh, et al. "Deep reinforcement learning at the edge of the statistical precipice." Advances in neural information processing systems 34 (2021): 29304-29320.

**Summary Of The Paper:**

This paper introduces a position-invariant regularizer to remove the absolute information from features while preserving the robust relational information. The effectiveness of the proposed method in improving generalization is evaluated on MAGICAL, Procgen benchmark and a real-world robot manipulation problem.

**Summary Of The Review:**

Overall, I think this paper is ok but requires some improvements to meet the acceptance bar.

---

> ### Author Response · Authors · 2022-11-12
> **Thanks for the review!**
>
> Thanks for the review! We address your concerns as follows.
>
> > 1. The experiments on Procgen are a little bit limited. It would be better to evaluate the proposed method on all 16 games.
>
> Thanks for the suggestion. We have added some more results in the text (so the experiments cover the environments used in the study [2]) and our method can still achieve the best results. Since we are more concerned about the manipulation-style tasks, we did not fully consider all the Procgen tasks in the original version and we are sorry for this.
>
> > 2. Apart from BC, does the position-invariant regularizer also work for other (maybe a SOTA one) imitation learning methods?
>
> We hypothesize that the proposed position-invariant regularizer may also work for other methods as it is a simple add-on to the policy network. We leave the investigation to future work.
>
> > 3. Compared to the baselines, how much extra cost (computation, time) does the adversarial training bring?
>
> The extra computation mainly comes from the adversarial procedure. In our experiments, we find this will make an epoch have about 2-4x more computation. But this is still practical since BC is a fast algorithm. For our robotics experiments, the training only takes about 30-40 minutes.
>
> > 4. I suggest the authors using the metrics recommended in [1] to compare different methods.
>
> This is a good suggestion. We will try to update accordingly.
>
> References
>
> [1] Agarwal, Rishabh, et al. "Deep reinforcement learning at the edge of the statistical precipice." Advances in neural information processing systems 34 (2021): 29304-29320.
>
> [2] Xin Chen, et al. "An empirical investigation of representation learning for imitation". In Neural Information Processing Systems Datasets and Benchmarks Track (NeurIPS), 2021.

---

> > ### Comment · Reviewer_khtv · 2022-11-18
> > **Thank you for the response**
> >
> > I appreciate the authors' efforts in the response. However, it does not fully resolve my concerns.
> >
> > * Can the authors explain why the selected 8 Procgen games are manipulation-style tasks while the other 8 games are not?
> >
> > * As the authors said, the proposed method POINT is a simple add-on. That is why I expect to see if it offers improvement across different imitation methods.
> >
> > * For a 2~4x increase in computation, I think it is reasonable to ask whether it is indeed advantageous to use this method than simply using a larger network. I am not very familiar with imitation learning, but in Procgen we can easily trade computation cost (by increasing the number of channels in conv layers) for higher performance.
> >
> > * Please consider reporting results in the metrics recommended in [1].
> >
> > [1] Agarwal, Rishabh, et al. "Deep reinforcement learning at the edge of the statistical precipice." Advances in neural information processing systems 34 (2021): 29304-29320.

---

### Official Review · Reviewer_efvU · 2022-10-22

**Confidence:** 3
**Correctness:** 3
**Technical Novelty And Significance:** 3
**Empirical Novelty And Significance:** 2
**Recommendation:** 6

**Clarity, Quality, Novelty And Reproducibility:**

The paper is clearly written, thorough in its experimentation, and the provided code should make the experiments reproducible.

**Strength And Weaknesses:**

Strengths:
- interesting approach to solving the problem, and solid demonstration that solving this particular 'structural generalization' problem matters to improving BC policies.
- paper is thorough, with a very good description of the method, and a sensible set of experimental results that provide convincing evidence.
- code is provided
- real-world experiments are provided, which greatly enhances the value of the experimental validation for robotics problems.

Weaknesses:
- The language used in the paper is sometimes getting in the way of clarity. Calling relative position invariance 'structural generalization' and expressing everything under that lens makes things unnecessarily difficult to read at first glance.
[edit post rebuttal: addressed]
- The level of complexity thrown at the problem in order to (partially) solve it is a definite concern: first you need to provide a saliency model, which are notoriously brittle to all but the simplest real-world images, and then you need to learn a complex discriminator model on top of it. This likely makes the approach prohibitive in practice, and means its usefulness is limited to being one purely academic step in addressing the problem.
[edit post-rebuttal: Acknowledging that http://gradcam.cloudcv.org/ was shown to be robust to real-world scenarios in prior works. Validation of its use in real-world downstream tasks is still missing (not a huge concern) and excessive complexity-to-outcomes ratio argument still stands]
- A number of works (https://transporternets.github.io/ comes to mind) bypass the issue of learning position-invariant representations by decoupling the policy representation from the visual representation learning, and having the policy only reason about the 'delta' between visual representations, while still enabling end-to-end differentiability of the architecture. I would have loved to see more discussion of such approaches.

**Summary Of The Paper:**

This paper proposes a method for learning 2D visual representations which explicitly discourage the model from memorizing absolute positions of objects on the scene, while still enabling the model to reason about their relative arrangement in order to derive action policies.

**Summary Of The Review:**

I am a bit torn. While I can't find any major flaw in the paper, it remains the case that it's the scientific equivalent of using a nuclear bomb to hammer in a nail, and only succeeding in inserting it halfway. There is scientific relevance to such study, but little that would generalize beyond the narrow setting of the problem. My confidence level in this assessment is relatively low.
[edit-post rebuttal: some concerns well-addressed. 5->6]

---

> ### Author Response · Authors · 2022-11-12
> **Thanks for the review!**
>
> Thanks for the interesting review! We are glad that you find our paper interesting and our evaluation solid. We address your concerns as follows.
>
> > 1. The language used in the paper is sometimes getting in the way of clarity. Calling relative position invariance 'structural generalization' and expressing everything under that lens makes things unnecessarily difficult to read at first glance.
>
> Thanks for the suggestion. We change the ‘structural generalization’ to ‘spatial generalization’ to reduce the confusion. We welcome for suggestions on this.
>
> > 2. The level of complexity thrown at the problem in order to (partially) solve it is a definite concern: first you need to provide a saliency model, which is notoriously brittle to all but the simplest real-world images, and then you need to learn a complex discriminator model on top of it. This likely makes the approach prohibitive in practice, and means its usefulness is limited to being one purely academic step in addressing the problem.
>
> This is an interesting concern. Fortunately, GradCAM is not brittle to real-world images. It is one of the most important tools in the CV community. Moreover, the ultimate objective of our method is to ensure that the feature of the policy does not contain the absolute position information of its attentive regions (which does not necessarily correspond to the most salient object). GradCAM can exactly compute the attentive region here.
>
> > 3. A number of works (https://transporternets.github.io/ comes to mind) bypass the issue of learning position-invariant representations by decoupling the policy representation from the visual representation learning, and having the policy only reason about the 'delta' between visual representations, while still enabling end-to-end differentiability of the architecture. I would have loved to see more discussion of such approaches.
>
> Thanks for this interesting question and pointing out these interesting works! We find the existing relevant works in robotics usually make somewhat stronger assumptions about the image based on specific application scenarios. Take the transporter as an example, it assumes that we can extract an orthographic projection image of the scene from an RGBD camera. We do not make this kind of assumption about the observation. Hence, our method can be used in a more general task setting. We will also add these works to the main text discussion.

---

> > ### Comment · Reviewer_efvU · 2022-11-14
> > **Thank you for the response.**
> >
> > Updated review accordingly.

---

### Official Review · Reviewer_xoNn · 2022-10-23

**Confidence:** 4
**Correctness:** 3
**Technical Novelty And Significance:** 2
**Empirical Novelty And Significance:** 2
**Recommendation:** 3

**Clarity, Quality, Novelty And Reproducibility:**

The paper is clearly written and introduces a novel approach. However, it lacks some important qualities in terms of scoping and experimental evaluation. It also lacks quite a few details important for reproducability, in particular with regards to the robot experiments.

**Strength And Weaknesses:**

# Strengths
- the approach is well-explained and easy to follow
- it is demonstrated across three different environments, including a real robot
- Fig 2 makes it quite easy to understand the approach
- the discussion of related work is comprehensive

# Weaknesses

There are several major concerns regarding the problem formulation, approach and experimental evaluation:

(A) **limited definition of structural generalization**: The paper defines structural generalization as generalization to scenes with different absolute positions of objects (and thus argues that in order to allow policies to perform structural generalization, we need to remove absolute position information form the learned representations). However, this is a very limited definition of structural generalization: there are many other aspects of such generalization, e.g. generalization to new instances of objects, new distractors, new scene layouts (that also involve relative location changes), more instances of the objects (eg pick up two cups after being trained on picking up one cup). As a result the scoping of the paper seems overly general for the actual invariances that the model introduces. Alternative approaches mentioned in the related work, e.g. the object centric approach by Zhou et al. 2022, can achieve a broader range of the aforementioned structural policy generalization.

(B) **objective can encourage information-less representations**: The introduced objective uses a discriminator to decrease mutual information between the regions of attention of the policy (computed via GradCAM) and the learned representation. There are two ways to reduce this mutual information: by removing position information from the representation or by uniformly attending to all parts of the image (so that the GradCAM attention visualization is uninformative). The latter solution is undesirable and does not align with the intuition of the proposed method, but it seems that the objective does not explicitly encourage the first over the second solution.

(C) **experiments don’t describe how structural generalization is tested**: The description of the experimental evaluation does not explicitly outline how structural generalization is tested. What is the structural difference between the training demonstrations and the test tasks for the different environments? This information is crucial and without it understanding the experimental results is hard. (For example in the MAGICAL environment experiments the demonstrations already use randomized object positions, so it is hard to understand what unseen structural generalization is required at test time)

(D) **missing baselines: random crop data augmentation**: The proposed method is an approach for inducing a certain invariance in the policy (absolute position invariance). As mentioned in the related work, an alternative for inducing invariances is data augmentation. While the experimental evaluation compares to multiple augmentation methods, it surprisingly omits random image cropping. This data augmentation has been shown to be the most effective for visual policy learning in prior work (DrQ, RAD) and also aims to induce positional invariance. At the same time it is substantially easier than the proposed approach. Thus, a comparison to this baseline and a qualitative comparison of the learned representation is crucial.


There is also a number of smaller points of critique:

(E) qualitative analysis does not investigate positional information in representation: the presented qualitative analysis (Fig 4) does not actually investigate the learned representation or the core property the paper tries to induce in it, positional invariance. It merely shows that the policy learns to attend to the relevant objects, but I would expect any functioning policy to show such patterns. Instead, some measure of the mutual information between the learned representation and the absolute object position should be reported and compared to baselines.

(F) dropout model attends to distractor, but not relevant objects: following the previous point, it is unclear to me why the dropout policy in Fig 4 would learn to attend to the distractor. I would expect any policy that properly learns to solve the task to attend to the relevant objects, not the distractor. The main qualitative difference between the baselines and the proposed approach should be in the degree to which the learned representation contains absolute position information. This could suggest that the baselines had some training issues unrelated to positional information.

(G) no ablation studies: the paper does not perform ablation studies of the proposed approach — for example it could be good to ablate the simplified vs full loss function (eq. 3 vs 4) and the introduced target network.

(H) not enough details about the robot experiment: the description of the robot experiment is too short, not enough details are provided eg about the action space of the agent, about why only the workspace camera encoder is regularized or how structural generalization is tested between training and test (see point (C) above).

# Questions

- How would the method perform for tasks with moving cameras where the 2D GradCAM projection might not accurately reflect the 3D position of the attended objects?

- Why does the proposed regularization, which aims to remove absolute object position, make the policy more robust to distractors? This is not intuitively clear to me.

- How would the method perform in tasks where absolute object position is important?

**Summary Of The Paper:**

The paper proposes an approach for inducing invariance to absolute object positions in visual imitation learning policies by determining the policy’s regions of attention in the input image and using a discriminator to ensure the learned visual representations do not encode information about their position. In experiments on abstract block pushing tasks, ProcGen and a real robot block pushing task the introduced regularization leads to better policy performance than a number of previous regularizers.

**Summary Of The Review:**

I have a number of concerns with regards to the proposed method and it’s experimental evaluation. In particular, I believe the paper is too broadly scoped for the proposed method, its not clearly explained how the experiments test the desired generalization capabilities and vital baselines are missing. Thus, I cannot recommend acceptance of the paper.

---

> ### Author Response · Authors · 2022-11-12
> **Thanks for the review! Response Part 1**
>
> Thanks for your thoughtful and constructive review! We address your concerns as follows.
>
> > (A) limited definition of structural generalization. The paper defines structural generalization as generalization to scenes with different absolute positions of objects (and thus argues that in order to allow policies to perform structural generalization, we need to remove absolute position information form the learned representations). However, this is a very limited definition of structural generalization: there are many other aspects of such generalization, e.g. generalization to new instances of objects, new distractors, new scene layouts (that also involve relative location changes), more instances of the objects (eg pick up two cups after being trained on picking up one cup). As a result the scoping of the paper seems overly general for the actual invariances that the model introduces. Alternative approaches mentioned in the related work, e.g. the object centric approach by Zhou et al. 2022, can achieve a broader range of the aforementioned structural policy generalization.
>
> Great suggestion. We agree that the term ‘structural’ sounds a bit broad. Therefore, we change the description to ‘spatial generalization’ and update this term in the text (including the abstract and title) accordingly. Now, spatial generalization means the generalization to the different, unseen setups of object locations. We welcome any further suggestions on this.
>
> > (B) objective can encourage information-less representations. The introduced objective uses a discriminator to decrease mutual information between the regions of attention of the policy (computed via GradCAM) and the learned representation. There are two ways to reduce this mutual information: by removing position information from the representation or by uniformly attending to all parts of the image (so that the GradCAM attention visualization is uninformative). The latter solution is undesirable and does not align with the intuition of the proposed method, but it seems that the objective does not explicitly encourage the first over the second solution.
>
> This is an interesting concern. Fortunately, the latter case does not happen in practice. We explain the process as follows.  During experiments, we find that the GradCAM visualization is only uninformative (uniform) when the BC loss is high. As long as we select an appropriate small $\lambda$, the model will keep minimizing the BC loss. As a result, the GradCAM activation will shrink to semantic objects or regions (which is highly non-uniform in practice).
>
> > (C) experiments don’t describe how structural generalization is tested. The description of the experimental evaluation does not explicitly outline how structural generalization is tested. What is the structural difference between the training demonstrations and the test tasks for the different environments? This information is crucial and without it understanding the experimental results is hard. (For example in the MAGICAL environment experiments the demonstrations already use randomized object positions, so it is hard to understand what unseen structural generalization is required at test time)
>
> A: Thanks for the question. We follow an IID distribution evaluation protocol as mentioned in [1]. That is, the training distribution and the testing distribution are the same (randomly sample object/target/ego poses) but note that the testing variation can be unseen. This is a common setup in recent robot learning work [4]. For our setting, even if the training distribution and the testing distribution are the same, there are thousands of possible object setups, which is far more exceeding the number of provided demonstrations.
>
> > (D) missing baselines: random crop data augmentation. The proposed method is an approach for inducing a certain invariance in the policy (absolute position invariance). As mentioned in the related work, an alternative for inducing invariances is data augmentation. While the experimental evaluation compares to multiple augmentation methods, it surprisingly omits random image cropping. This data augmentation has been shown to be the most effective for visual policy learning in prior work (DrQ, RAD) and also aims to induce positional invariance. At the same time it is substantially easier than the proposed approach. Thus, a comparison to this baseline and a qualitative comparison of the learned representation is crucial.
>
> A: Sorry for not including this in the text. We have added the result. We found that it does not help that much, though it is indeed a strong baseline. Some recent work [2] in imitation learning also identifies a similar result.

---

> > ### Author Response · Authors · 2022-11-12
> > **Thanks for the review! Response Part 2**
> >
> > > (E) qualitative analysis does not investigate positional information in representation.
> >
> > Thanks for the suggestion. We measure the suggested mutual information using the MINE estimator [3] and find that our method contains 60% fewer bits of absolute information compared with the baseline. We have incorporated the result into the qualitative results.
> >
> > > (F) dropout model attends to distractor, but not relevant objects.
> >
> > This is an interesting and important concern. Previous studies also find that the BC model can attend to task-irrelevant objects, such as the background (Figure 4 of [2]). Therefore, it may not be so surprising that the dropout model can attend to the distractor.
> >
> > > (G) no ablation studies: the paper does not perform ablation studies of the proposed approach — for example it could be good to ablate the simplified vs full loss function (eq. 3 vs 4) and the introduced target network.
> >
> > Thanks for the suggestion. We will add the results. When we use equation 3 (full loss), the performance drop by 18% on MAGICAL on average. When we do not use target network, the performance drops by 11% on MAGICAL on average.
> >
> > > (H) not enough details about the robot experiment: the description of the robot experiment is too short, not enough details are provided eg about the action space of the agent, about why only the workspace camera encoder is regularized or how structural generalization is tested between training and test (see point (C) above).
> >
> > Thanks for the suggestion and we have updated the text. Note that this is a pick and place task rather than a pushing task. We use a discretized action space, so it will be easier for humans to collect the demonstration via the keyboard. There are 8 actions in total, (x/y/z) move (backward/forward 2cm), and gripper open/close 1cm. We only consider regularizing the workspace camera encoder for the following reason. Since the gripper camera can only observe a very limited area of the workspace in our case and the hand observation is always at the center, the spatial generalization issue is minor for the gripper camera. We find it does not help much to regularize it. The training & testing protocol is the same as (C) above.
> >
> > > Q1: How would the method perform for tasks with moving cameras where the 2D GradCAM projection might not accurately reflect the 3D position of the attended objects?
> >
> > This is an interesting question. However, recent works usually use fixed workspace cameras. Part of the ProcGen experiments in this paper (like CoinRun) involves the ego-centric moving camera, but the structure is two-dimensional. We will examine the case of using a moving camera in 3D space in the future.
> >
> > > Q2. Why does the proposed regularization, which aims to remove absolute object position, make the policy more robust to distractors? This is not intuitively clear to me.
> >
> > Good question. We can think about this from a feature selection perspective. If we do not remove the absolute position, then there can exist many features correlated to the absolute position of each object. In this case, the model has more chance to exploit non-robust features that correlated to the absolute information of the distractor (especially when we do not have sufficient many data).
> >
> > > Q3. How would the method perform in tasks where absolute object position is important?
> >
> > This is an interesting concern. However, most of the manipulation tasks involve the interaction between the robot and the objects, in which understanding the relation information between them is always essential. If a task is highly dependent on absolute information, such as whether an object is at a fixed position, then it may be quite easy to solve either with rule-based or learning-based methods.
> >
> > References
> >
> > [1] Robert Kirk, et al. A survey of generalisation in deep reinforcement learning. arXiv preprint arXiv:2111.09794, 2021
> >
> > [2] Xin Chen, et al. An empirical investigation of representation learning for imitation. In Neural Information Processing Systems Datasets and Benchmarks Track (NeurIPS), 2021
> >
> > [3] Mohamed Ishmael Belghazi, et al. Mutual information neural estimation. In International Conference on Machine Learning (ICML), 2018
> >
> > [4] Mohit Shridhar, et al. Perceiver-actor: A multi-task transformer for robotic manipulation. In Conference on Robot Learning (CoRL), 2022

---

> > > ### Comment · Reviewer_xoNn · 2022-11-24
> > > **Thanks for your response!**
> > >
> > > Thanks for taking the time to answer my review. It is great to see that random-crop augmentation was included as a baseline and I think the MI evaluation makes the analysis of the learned representation stronger!
> > >
> > > My concern with the evaluation setup remains however: in the rebuttal the authors mention that they train and test **on the same (position) distribution** (I understand that individual positions in the test set were not seen in the training set). Yet, a paper on structural generalization that does not explicitly analyze the method under the lens of generalization seems incomplete. I would suggest the following, more convincing experimental setup: have a rather narrow training position distribution, then have multiple test distributions with increasing diversity in positions. Show that on the narrow test distributions the baselines work roughly as well as the proposed method (this shows the baselines are strong & implemented correctly), then show that as we increase the diversity in positions the proposed method deteriorates in performance much less than any of the baselines (ie generalizes better). Such a setup would much better support the claimed contribution of the paper than the current evaluation.
> > >
> > > Also, I am not convinced by the argument why the dropout policy attends to distractors: if the policy properly learned to solve the task, I would expect it to at least *also* attend to the correct objects. I also do think that adding some discussion about moving cameras to the paper would be valuable — while many current table-top manipulation works indeed use fixed 3rd person cameras, any mobile manipulation / navigation task *requires* a moving camera so a method that only works on static cameras would be severely limiting.
> > >
> > > Thanks again for taking the time for the rebuttal, I think it strengthened the paper. However, I do think the experimental evaluation is not sufficient for a paper that is specifically geared towards improving (positional) generalization (see my suggestion for a more convincing evaluation above). Thus, I still do not recommend acceptance.

---

### Decision · Program_Chairs · 2023-01-20

**Decision:**

Reject

**Justification For Why Not Higher Score:**

Explained in the meta-review

**Justification For Why Not Lower Score:**

-

**Metareview: Summary, Strengths And Weaknesses:**

This paper had received two negative reviews and one slightly positive one and was  heavily leaning towards the rejection. While the general idea was found interesting, even the slightly positive review had a mixed touch to it and reviewer efvU described the method as overly complex for a simple problem, which was not entirely solved. While the authors could answer and clear up some minor issues, the main problems remained:

- Limited definition of structural generalization and also identical distribution between training and testing. The answers the authors provided on this important point were considered as unconvincing.
- Missing baselines and ablation studies.
- Lack of details.
- Overly complex formulation to address a quite simple problem.

The AC considers that the paper is not yet ready and recommends rejection.

**Summary Of Ac-Reviewer Meeting:**

No meeting was held